
# A study of the approaches used to retrieve aerosol extinction, as applied to limb observations made by OSIRIS and SCIAMACHY

Landon A. Rieger[1], Elizaveta P. Malinina[2], Alexei V. Rozanov[2], John P. Burrows[2], Adam E. Bourassa[1], and Doug A. Degenstein[1]

[1]Institute of Space and Atmospheric Studies, University of Saskatchewan, Saskatoon, Canada
[2]Institute of Environmental Physcis, University of Bremen, Bremen, Germany

*Correspondence to:* Landon Rieger (landon.rieger@usask.ca)

**Abstract.** Limb scatter instruments in the UV-Vis spectral range have provided longterm global records of stratospheric aerosol extinction important for climate records and modelling. While comparisons with occultation instruments show generally good agreement, the source and magnitude of the biases arising from retrieval assumptions, approximations in the radiative transfer modelling, and inversion techniques has not been thoroughly characterized. This paper explores the biases between SCIA-

MACHY v1.4 , OSIRIS v5.07 and SAGE II v7.00 aerosol extinctions through a series of coincident comparisons as well as simulation and retrieval studies to investigate the cause and magnitude of the various systematic differences. The effect of a priori profiles, particle size assumptions, radiative transfer modelling, inversion techniques, and the different satellite datasets are explored. It is found that the assumed a priori profile can have a large effect near the normalization point, as well as systematic influence at lower altitudes. The error due to particle size assumptions is relatively small when averaged over a range

of scattering angles, but individual errors depend on the particular scattering angle, particle size and measurement vector definition. Differences due to radiative transfer modelling introduce differences between the retrieved products of less than 10% on average, but can introduce vertical structure. The combination of the different scenario simulations and the application of both algorithms to both datasets enable the origin of some of the systematic features such as high altitude differences when compared to SAGE II to be explained.

*Copyright statement.*

## 1 Introduction

Stratospheric aerosols play an import role in several atmospheric processes, including radiative forcing and ozone depletion. For decades, monitoring of stratospheric aerosols from satellite observations was largely the domain of occultation instruments such as SAGE II. However, since the 2000's aerosol extinction has been retrieved from limb scatter instruments such as the

Optical Spectrograph and InfraRed Imaging System (OSIRIS) (Llewellyn et al., 2004; Bourassa et al., 2012, and references therein), the SCanning Imaging Absorption spectroMeter for Atmospheric CHartographY (SCIAMACHY) (Burrows et al., 1995; Bovensmann et al., 1999; von Savigny et al., 2015, and references therein) and the Ozone Mapping and Profile Suite -



Limb Profiler (OMPS-LP) (Flynn et al., 2006; Loughman et al., 2017). While limb scatter provides greatly improved global coverage over occultation satellites, it requires additional assumptions and computationally expensive forward models to perform the inversions. Despite the difficulties, comparisons between limb scatter and occultation measurements generally agree favourably with mean biases in the 10-15% range during volcanically quiescent periods. While this is the average case, biases

at certain latitudes and altitudes can be considerably larger. Additionally, biases after 2005 have not been well characterized due to the lack of baseline occultation measurements with which to compare.

This paper investigates the cause of the biases between the OSIRIS and SCIAMACHY aerosol extinction retrievals using comparisons with SAGE II and a series of simulation studies. The two limb-scattering instruments and the inversion techniques are described in Section 2. Also introduced here is the new version 1.4 SCIAMACHY aerosol extinction product used in this

work. Initially, a triple comparison between OSIRIS, SCIAMACHY and SAGE II is performed in section 3. As there was very little volcanic influence on the stratospheric aerosol load during the overlap period, this serves as a baseline for the agreement seen between the limb scatter and occultation aerosol records during volcanically quiescent times, and motivates the investigation of error sources. Section 4 discusses the magnitudes of the errors that are expected from the assumptions in the OSIRIS and SCIAMACHY retrievals and radiative transfer models through a series of simulation studies. Section 5 applies

the IUP and USask retrievals to both datasets to investigate differences due to the inversion techniques and radiance products. Lastly, conclusions and recommendations are discussed in Section 6

## 2    The Aerosol Retrievals

Generally, aerosol extinction retrievals for OSIRIS, SCIAMACHY, and OMPS-LP limb scattering instruments proceed in a similar fashion. First, radiance profiles at one or more wavelengths are used to construct a single measurement vector as a

function of altitude. As this provides only one piece of information at each altitude, aerosol extinction is typically chosen as the retrieved quantity, although this is not the only possibility. However, extinction is the natural quantity retrieved from occultation instruments, and allows for continuation of this historical record. Ideally, the measurement vector would be dependent only on the desired aerosol extinction parameter, but in practice is also affected by the surface albedo, atmospheric density, and aerosol optical properties including particle size, shape and composition. Typically, an effective Lambertian surface reflectivity

is retrieved concurrently with the aerosol extinction, while the atmospheric density and optical properties are assumed using external information. Although atmospheric density is provided at high resolution by ECWMF or MERRA, data on aerosol optical properties is much sparser, and a notable limitation in the current retrievals.

Although particle size information has been retrieved from limb instruments in the past with OMPS-LP and OSIRIS (Rault and Loughman, 2013; Rieger et al., 2014) and more recently with SCIAMACHY (Malinina et al., 2017), the standard opera-

tional products remain as extinction-only for the OSIRIS and OMPS-LP aerosol products. These extinction products have been used in numerous studies and continue to contribute to the stratospheric aerosol record (Kremser et al., 2016; Thomason et al., 2017), highlighting the importance of accurately characterizing not only precision, but also biases in the current operational retrievals.



## 2.1 OSIRIS v5.07

The Optical Spectrograph and InfraRed Imaging System (OSIRIS) was launched in 2001 aboard the Odin spacecraft (Llewellyn et al., 2004). The spectrograph produces limb scattered radiance profiles from 280 to 810 nm with typical sampling every 2 km and a vertical resolution of 1 km, and an altitude range from 7 to 75 km. Odin is in a near terminator orbit with an equatorial

crossing time of approximately 6 A.M. on the descending node, providing limb measurements with a limited range of viewing geometries. Typically, solar scattering angles vary between $60°$ and $120°$ with the largest values occurring in the tropics, and little correlation between the mean scattering angle and latitude. The OSIRIS measurements have been used in the inversions of multiple species with products now spanning over 15 years (McLinden et al., 2012). The inversions use the SASKTRAN radiative transfer model (Bourassa et al., 2008; Zawada et al., 2015) and a multiplicative algebraic reconstruction technique

(MART) to retrieve ozone, $NO_2$ and aerosol extinction at 750 nm. This paper uses the OSIRIS v5.07 aerosol data product retrieved with the algorithm discussed in Bourassa et al. (2007, 2012), which simplifies to the Chahine inversion technique (Chahine, 1970) for the choice of tangent altitude weighting factors in the aerosol-specific portion of the MART retrieval. This algorithm will be referred to as the USask retrieval in this paper. For the radiative transfer modelling, a unimodal lognormal distribution is assumed with median radius, $r_g$ of 80 nm and distribution width, $\sigma_g$, of 1.6 as defined in Equation 4. Mie theory

is used to calculate the aerosol scattering properties with a refractive index from Palmer and Williams (1975) assuming a 75% concentration of $H_2SO_4$ and 25% $H_2O$. The USask measurement vector is defined as,

$$y_{jk} = \ln\left(\frac{I(\lambda_k, j)}{I(\lambda_{\mathrm{ref}}, j)}\right) - \frac{1}{N}\sum_{j_{\mathrm{ref}}=m}^{m+N} \ln\left(\frac{I(\lambda_k, j_{\mathrm{ref}})}{I(\lambda_{\mathrm{ref}}, j_{\mathrm{ref}})}\right), \tag{1}$$

where the measurement vector, $y_{jk}$ at wavelength $k$ and altitude $j$ is the radiance, $I$, normalized by a reference altitude, $j_{\mathrm{ref}}$, and shorter wavelength, $\lambda_{\mathrm{ref}}$ that is less sensitive to aerosols. The reference altitude is chosen as the point, or points, where the

total signal from both aerosol and stray light is at a minimum, leading to a normalization that varies scan-to-scan. This typically produces reference altitudes between 25 and 40 km with lower altitudes near the poles. To improve the convergence speed of the relaxation technique a modelled measurement vector assuming a molecular atmosphere, is also used as a normalization. As this acts as a constant offset, it does not affect the sensitivity of the measurement vector to aerosols. For the USask retrieval, 750 nm is used as the long wavelength, $\lambda_k$, and 470 nm is used as the reference, $\lambda_{\mathrm{ref}}$.

## 2.2 SCIAMACHY v1.4

SCIAMACHY, the SCanning Imaging Absorption spectroMeter for Atmospheric CHartographY instrument (Burrows et al., 1995; Bovensmann et al., 1999) was a national contribution to the payload on ESA's Envisat Satellite, which was launched in March 2002. Envisat was placed in a sun-synchronous orbit at 800 km altitude with an equatorial crossing time of 10:00 A.M. on the descending node. This results in solar scattering angles ranging from $30°$ in the high Northern latitudes to $150°$ in the

high Southern latitudes with a strong latitudinal dependence. SCIAMACHY operation started in August 2002 and ended with a sudden loss of communication with the Envisat satellite in April 2012. SCIAMACHY performed measurements in 8 spectral channels covering a wide spectral range from 214 to 2380 nm with a resolution varying from 0.2 to 1.5 nm. During its mission,





SCIAMACHY measured the solar radiation in nadir, limb-scatter and solar/lunar occultation geometries and provided daily measurements of the solar spectral irradiance that have been used to retrieve a variety of species including aerosols, clouds, ozone, BrO, $NO_2$, and water vapour. For this study stratospheric aerosol retrievals are performed using the data from the limb-scatter viewing geometry, where measurements are provided every 3.3 km with a vertical resolution of 2.6 km in the altitude range from approximately 0 to 100 km.

The stratospheric aerosol extinction retrieval algorithm used in this study is an updated version of the algorithm described by von Savigny et al. (2015) and Ernst et al. (2012). The SCIAMACHY v1.4 retrievals, herein referred to as the IUP retrievals, use the newer version 8 SCIAMACHY Level 1 radiance data. Atmospheric pressure and temperature background profiles from ECMWF (European Center for Medium-Range Weather Forecasts) operational analysis data from the specific date, time and location of each SCIAMACHY limb measurement are used. In comparison to the previous version of the algorithm (von Savigny et al., 2015; Ernst et al., 2012) and the USask retrieval algorithm, the updated v1.4 algorithm drops the shorter, 470 nm wavelength normalization to reduce the uncertainties related to measurement noise and lower sensitivity to aerosols. The new measurement vector is given by,

$$y_{jk} = \ln\left(I(\lambda_k, j)\right) - \ln\left(I(\lambda_k, j_{\text{ref}})\right). \tag{2}$$

To reduce noise on the measurements, all measured wavelengths within $\pm 2$ nm of $\lambda_k$ are used in the retrieval. For the v1.4 extinction product the aerosol profiles are retrieved at 750 nm. The retrieval uses measurements in the altitude range from around 12 to 35 km (depending on the latitude and season) with a reference tangent altitude of about 38 km. The v1.4 aerosol extinction retrieval is performed on the measurement altitude grid, and the values below and above the retrieval range are fixed to the a priori. Effective Lambertian albedo of the underlying surface is concurrently retrieved based on the limb radiances near the reference tangent height to reduce the influence of clouds below the field of view, although clouds within the field of view remain an issue. To reduce their impact extinction values greater than 0.001 km$^{-1}$ are considered cloud contaminated and filtered after the retrieval is performed. To solve the inverse problem an iterative regularized inversion approach similar to that described by Rodgers (2000) is used. As in Ernst et al. (2012) it is assumed that the errors are uncorrelated, and the noise covariance matrix is chosen to be diagonal. Signal-to-noise ratio is set to 200 for all tangent heights. For the a priori covariance matrix the non-diagonal elements drop off exponentially with a correlation radius of 3.3 km and the diagonal elements correspond to a relative standard deviation of one.

Forward modelling, as well as retrievals, are done using the radiative transfer model with the retrieval code SCIATRAN-3.7 (Rozanov et al., 2014). The scattering phase functions are calculated using Mie scattering theory, assuming spherical sulphate aerosol particles with a unimodal, lognormal size distribution. The refractive indices are calculated using the OPAC database (Hess et al., 1998). The stratospheric aerosol parameters are defined from 12 to 46 km, where it is assumed to consist of sulfuric droplets with 0% relative humidity in the surrounding atmosphere. To exclude additional uncertainty associated with the aerosol particle size distribution, the same distribution parameters as for OSIRIS retrieval are used ($r_g$=80 nm , $\sigma_g$=1.6). We note that both the parameters used in the OSIRIS retrieval and those used by von Savigny et al. (2015) ($r_g$=110 nm , $\sigma_g$=1.37) are consistent with in situ observations (Deshler et al., 2003), and there is no evidence that any one should be preferred.





## 3   Coincident Comparisons with SAGE II

The Stratospheric Aerosol and Gas Experiment (SAGE) II was launched in 1985 and operated until November 2005, providing one of the longest, continuous records of stratospheric aerosols. As an occultation instrument, the SAGE II aerosol retrieval is insensitive to many of the assumptions required in the limb scatter retrievals, making for a robust, independent comparison. This

work uses the version 7.00 SAGE II aerosol extinction data at 525 and 1020 nm (Damadeo et al., 2013). Several improvements have been made since version 6.2 that have resulted in aerosol extinction decreasing more quickly at higher altitudes. As both the OSIRIS and SCIAMACHY aerosol products are produced at 750 nm, the SAGE II data is interpolated to this wavelength using the Ångström coefficient derived from the 525 and 1020 nm channels. Although this is not a perfect conversion, as the wavelength dependence is not strictly linear in log-wavelength log-extinction space, the error is generally limited to less than

10% for most particle sizes (Rieger et al., 2015). To test agreement between the three instruments a coincident comparison is performed when all instruments have collocated measurements. Measurements are used when OSIRIS and SCIAMACHY observations are within $\pm 5°$ latitude, $\pm 20°$ longitude, and $\pm 24$ hours of the SAGE II tangent point. To minimize the impact of clouds in the analysis extinction values greater than $0.0025 \, \text{km}^{-1}$ have been excluded. Due to the relatively eruption-free period of this comparison this has minimal effect on the comparisons removing approximately 3% of scans above 15 km and none

above 20 km. This criterion provides 2580 coincident measurements between 2002 and 2005, when all three instruments were operating. The comparison is broken into $20°$ latitude bins to better distinguish biases related to latitude and solar geometry conditions. Results are shown in Figure 1. In general, all instruments agree to within approximately 15% for most regions. Exceptions to this are at high altitudes and latitudes (such as Panels A, B and H) where both OSIRIS and SCIAMACHY retrieve lower values than SAGE II by up to 40% at 30 km. At latitudes above $40°N$ SCIAMACHY shows systematically

higher results than SAGE II for all altitudes below 30 km. This effect increases with latitude up to approximately 40% at the highest Northern latitudes, and is visible in panels G and H of Figure 1. Although the largest clouds have been removed, both limb scatter instruments are likely to still contain some cloud contamination near and below the tropopause and the differences compared to SAGE II show large standard deviations in these regions.

Several factors are expected to contribute to the differences between the aerosol extinction retrieved from the measurements

of the occultation and limb scatter instruments, as well as the different biases between OSIRIS and SCIAMACHY. Limb scatter inversions use complex forward models which are not identical in their assumptions or approaches. The inversions themselves also differ in several ways; with SCIAMACHY using a regularized inversion technique and OSIRIS using MART. A priori assumptions, such as the choice of aerosol particle size distributions and extinction profiles also affect the retrievals. The importance of these effects depends on the viewing geometry of the instrument. OSIRIS and SCIAMACHY have significantly

different viewing geometries as a result of the Envisat and Odin orbits. The following sections explore the significance of these different effects.



## 4  Simulation Study

To test the sensitivity of the retrievals to assumed parameters and retrieval settings a series of simulation studies is performed. The 2580 near coincident scans from the SAGE II comparison are used as the test cases. These scans cover the full range of OSIRIS and SCIAMACHY geometries. While these scans are limited to pre-2006, the simulations use a range of atmo-

spheric scenarios consistent with both background and volcanically perturbed conditions. Four factors are investigated in this study: the impact of different radiative transfer models, a priori extinction profile and particle size assumptions, and choice of measurement vectors.

### 4.1  Radiative Transfer Modelling

It is difficult to decouple the retrieval algorithms from the radiative transfer models entirely due to differences in languages,

input formats, and interfaces. However, differences between the IUP and USask retrievals due to the radiative transfer models can still be estimated by simulating measurements using one model, and retrieving with the other. For this test, the SASKTRAN radiative transfer model is used to generate radiances that simulate the OSIRIS measurements. These synthetic radiances are then used in the IUP retrieval which uses the SCIATRAN radiative transfer model. The same is then performed with the SCIA-TRAN simulated radiances and the USask retrieval using SASKTRAN, again on OSIRIS measurements. Although this is not

a test of 'correctness' of either model, nor a test of how well the radiative transfer models could agree, it provides an indication of the magnitude of differences that should be expected due to the configuration of the radiative transfer models as used in the retrievals. Figure 2 shows the differences in the modelled radiances and retrievals. Panel A shows the differences in the radiances at the 470 and 750 nm wavelengths. The radiances have systematic differences of approximately 5%, with SCIATRAN producing larger radiance values than SASKTRAN. Some of this difference is due to model resolution settings. SASKTRAN

simulations are performed at a higher vertical resolution of 1 km, and when both models use this higher resolution vertical grid agreement is improved to within 2-5%. However, because the IUP retrieval is performed on a 3.3 km grid, the higher resolution is not required for SCIAMACHY retrievals. Although the variation in radiances between the models can occasionally reach 15%, the normalizations used in the measurement vectors cancel much of the systematic differences. This can be seen in Panel B, where differences in the measurement vectors, computed using the two different models, are shown. In this panel the red

curve shows the percent difference between the IUP retrieval vectors defined in equation 2 when computed from SASKTRAN versus SCIATRAN radiances. The blue curve shows the same, expect computed using the USask measurement vector defini-tion from equation 1. The high altitude normalization used in the IUP retrieval decreases the differences between the models to less than 2% at most altitudes. If the short wavelength normalization is included the difference is larger, typically near 5%, since the wavelength dependent modelling differences vary more with altitude. How this difference translates to the retrieved

extinction is shown in Panel C. Here, the red curve shows the difference in the IUP retrieved extinction using SASKTRAN generated radiances compared to the true state. Similarly, the blue curve shows the same for USask retrieved extinction using SCIATRAN generated radiances. The IUP retrieval produces errors in the retrieved extinction less than 5% for most of the aerosol layer, with a standard deviation close to 5% as well. The larger differences in the USask measurement vector lead to



larger differences in the USask the retrieved extinction, although errors are still typically less than 10%. The exception to this is below 17 km and above 30 km where the sensitivity to aerosol is low, and therefore small changes in the radiative transfer cause large changes in the extinction. This highlights that the high altitude normalization is effective not only in minimizing errors due to uncertainties in unknown physical quantities such as albedo, but also in reducing errors due to model assumptions.

Conversely, the short wavelength normalization has the potential to introduce additional error if the radiative transfer model biases change with wavelength.

## 4.2  A priori Profiles

The effect of the a priori profile on the retrieval is an important consideration and one that has the potential to vary substantially between retrieval methods. Although the MART relaxation used in the USask retrieval has no regularization, and the IUP

retrieval is only weakly constrained by the a priori, the effect of the a priori at altitudes above the retrieval range can still play an important role. The aerosol here can couple to the lower altitudes due to the high altitude normalization of the measurement vectors. While this normalization has many benefits, it has the drawback of coupling the error at high altitudes to all altitudes below. The USask retrieval scales the a priori above the retrieval range, at each iteration of the retrieval to match the top retrieved value and thus avoid sharp discontinuities in the retrieved profile. Therefore, the absolute error above the retrieval

range depends on the shape of the a priori profile at and above the normalization and the retrieved aerosol just below the normalization. Conversely, the IUP extinction is fixed to the a priori value above the retrieval altitudes, so will depend less on the shape of the chosen a priori and more on the absolute value in the normalization range.

The effect of the a priori above the retrieval range is tested through a simulation study where the true high altitude aerosol profile (i.e. the input profile used to generate the synthetic measurements) differs from that assumed in the retrievals. For

this test an exponentially varying aerosol profile above 30 km is taken to be the truth. The slope of the exponential profile is then varied for each simulated OSIRIS and SCIAMACHY scan. The range of exponential profiles used as true states in the simulations is shown as the grey shaded region in Figure 3. The USask and IUP a priori values are shown as the blue and red lines respectively. The shape of the a priori profile below 30 km, as well as all other aerosol parameters such as particle size, are assumed correctly in the simulated retrievals to avoid introducing errors due to other retrieval parameters. The simulated data

was then used to retrieve the extinction profile using the USask and IUP retrievals under two conditions. First, both retrievals are initialized with the USask a priori profile, and second, both are initialized with the IUP a priori profile.

Figure 4 shows the relationship between errors at the reference altitude to errors lower in the profile for four cases. The top row shows results for the USask retrieval with the bottom row showing the IUP retrievals. The left column shows results when the USask a priori profile is used for the retrievals with the right column showing results when the larger IUP a priori is used.

The solid line shows a linear best fit to the data. Generally, if aerosol is overestimated in the normalization range, due to an a priori profile that decays too slowly with altitude, the aerosol is overestimated for the entire retrieval. This is because the modelled vector is normalized by too-large a value, decreasing the magnitude in the retrieval range; as a result, extra aerosol is added to compensate. The error in the retrieved aerosol is very well correlated with the error in the normalization range, with little dependence on whether the USask or IUP retrieval is used. This holds well for all geometries tested, and for both retrieval





algorithms. However, higher altitudes are more sensitive to aerosol loading, and so show a larger error in the retrieved profile for a similar absolute error in the a priori as the normalization altitude is increased. This can be seen in the larger sensitivity to a priori errors in the IUP retrieval, which uses a 38 km reference height, as opposed to the USask retrieval that used 35 km. The same error of $10^{-6}\,\mathrm{km}^{-1}$ at a normalization altitude of 38 km will cause approximately twice the error that it does at 35 km.

At low altitudes, less than approximately 14 km, the sensitivity to aerosol is very low and the retrievals no longer show a clear relationship between the retrieval error and the a priori error.

     The altitude dependence of the retrieved error, normalized by the error at 35 km is shown in Figure 5. We note that normalizing the IUP retrieval by the error at 35 km is not strictly correct as the reference altitude is at 38 km. However, this allows for a consistent comparison between the two algorithms, and due to the relatively linear nature of the error it is not expected to

introduce large biases. The retrieval error is smallest at around 22 km, where the aerosol loading is highest, and the measurement sensitivity is still quite good, with error increasing above and below this altitude. The error can also be estimated without simulating the full retrieval using the equation,

$$\delta\mathbf{k} = \mathbf{G}\delta\mathbf{y}, \tag{3}$$

where $\delta\mathbf{k}$ is the error in the retrieved extinction, $\mathbf{G}$ is the gain matrix or the sensitivity of the retrieved extinction to variations

in $y$, and $\delta\mathbf{y}$ is the error in the measurement vector. In this case, $\delta\mathbf{y}$ is the error in the measurement vector due to errors in the assumed aerosol at the normalization altitude and above. As the retrieval error is quite linear with respect to errors in the high altitude profile, $\delta\mathbf{y}$ in the retrieval range can be calculated directly from the Jacobian matrix, $\mathbf{K}$. This analysis as applied to the USask retrieval is plotted in Figure 5 as the dashed line. Agreement between the analytic method and simulation study is excellent over the full range of values tested. As $\mathbf{G}$ and $\mathbf{K}$ are typically readily available from the inversion method, this can

also be applied on an operational basis if estimates of the extinction error at the normalization point are known.

## 4.3   Particle Size

In the standard extinction retrievals the aerosol optical properties are not retrieved, and must therefore be assumed when retrieving extinction. Of primary importance in the IUP , USask and OMPS retrievals is the assumption of a fixed particle size. All three retrievals assume lognormal distributions that correspond to typical background conditions as measured by Deshler

et al. (2003), albeit with somewhat different lognormal parameters. The lognormal distribution used in the retrievals is given by the equation:

$$n(r) = \frac{N}{\sqrt{2\pi}\ln(\sigma_g)r}\exp\left(\frac{(\ln(r_g) - \ln(r))^2}{2\ln^2(\sigma_g)}\right), \tag{4}$$

where $r_g$ is the median radius, $\sigma_g$ the distribution width, and $N$ the number density. During background conditions the median radius is generally larger than 40 nm but less than 200 nm, depending on altitude. However, after volcanic eruptions, a second

mode of particles with median radii up to a few microns may be present, further complicating the analysis. The effect of this constant unimodal particle size assumption was estimated to a degree by Rieger et al. (2015), however a limited number of geometries and cases were tested. More recently, Loughman et al. (2017) estimated the impact of particle size assumptions



based on estimates of the phase function, but did not fully propagate the error through the retrievals. This work extends these previous analyses to additional conditions and geometries, and estimates the impact on the retrieved extinction.

To estimate errors due to particle size assumptions two sets of simulations are performed. First, a study to estimate errors in the retrieved extinctions during relatively quiescent periods is done, when only a fine mode of aerosols is present. For these simulations, the fine mode lognormal parameter profiles as measured by the OPC in Wyoming by Deshler et al. (2003) between 2001 and 2014 are used as inputs for the simulated data. This provides 44 unique particle size profiles. To avoid noise and high frequency oscillations the OPC profiles are smoothed to a vertical resolution of approximately 3 km. The extinction profile was set to twice that of the a priori assumption, with no change in the shape to avoid including a priori errors in this portion of the study. The second set of simulations covers conditions representative of those after volcanic eruptions, when an additional mode of larger particles is present. For this case, the smoothed coarse mode as measured by the OPC is also added to the true extinction profile. The number densities of the fine and coarse modes are set such that the coarse mode accounts for 70% of the total extinction. In each case, the coincident OSIRIS and SCIAMACHY scans were simulated using a random OPC particle size profile and a random albedo between zero and one as the true state. Figure 6 shows the range of median radii, widths, and Ångström exponents (calculated between 525 and 750 nm) used in the simulations, as well as the a priori values.

The standard USask algorithm was then used to retrieve extinction using the simulated data. The four left panels in Figure 7 show these results. Each panel shows the relative error in the retrieved extinction as a function of the true Ångström coefficient at 20 km. The colour of each point indicates the scattering angle of the measurement. Panels A and B show results for the fine-mode only simulations, while C and D show results from bimodal cases. Panels A and C shows results from OSIRIS geometries, and those from SCIAMACHY geometries are presented in panels B and D. These retrievals were then repeated using the USask algorithm but without the short wavelength normalization to determine its effect, with results in the right four panels. If only fine mode particles are included in the simulated atmosphere, the error in the retrieval can be well parameterized by the Ångström coefficient and the solar scattering angle of the measurement. When the Ångström coefficient is assumed correctly the error in the retrieval is less than 10%, nearly independent of the particular lognormal parameters. As the error in the Ångström coefficient increases, so does the error in the retrieval, up to 100% for OSIRIS geometries. For SCIAMACHY geometries the range of scattering angles and errors can be larger, due to larger variations in the aerosol phase function at extremely large and small angles. With a short wavelength normalization the retrievals show errors that are mostly symmetric about zero. While this will help to reduce biases over longer periods of time when a large range of scattering angles are sampled, seasonal biases are still to be expected as different scattering angles are sampled over the course of a year. Similarly, latitudinal biases are likely in the SCIAMACHY data as scattering angle depends strongly on latitude. Without a short wavelength normalization the general spread and shape of the errors is similar; however, the errors are not centered about zero with aerosol being overestimated more often than not. In this case, the error is minimized during forward scattering conditions when scattering angles are near 60°. When short wavelength normalization is used the error is at a minimum near 90°; subsequently the error for forward scatting geometries is increased, while it is decreased for backscattering geometries.

When coarse mode particles are included, the phase functions can vary more widely for a given Ångström coefficient, leading to less of a clear relationship in the retrieved error. This can be seen in panels C and D of Figure 7, where much





weaker correlation between the Ångström coefficient, solar scattering angle, and extinction error is visible. Even when the Ångström coefficient is assumed correctly, differences in the lognormal parameters can induce errors of 30% in the retrieval for OSIRIS geometries, and 50% for SCIAMACHY geometries. While the error is less correlated, errors are not systematically larger than during volcanically quiescent periods, but do have a tendency to introduce low biases in the retrieved results for

most geometries and particle sizes. Additionally, while backscatter can still have large biases, they are not as large at the extreme scattering angles as during fine mode only conditions. During bimodal conditions the error in both the normalized and non-normalized retrievals is comparable, except during strongly forward scattering conditions when the short wavelength normalization increases the error. In general, this shows that the short wavelength normalization is beneficial during background periods under backscattering conditions, but generally increases the error during forward scatter. Additionally, in forward

scatter both the 470 and 750 nm wavelengths are positively sensitive to aerosol, so the wavelength ratio will tend to decrease the sensitivity to aerosol and decrease the retrieved precision due to measurement noise as well.

## 5   Retrieval Study

In Section 4 the sensitivity to retrieval assumptions and radiative transfer modelling was estimated. In this section, we explore the applicability of the USask retrieval to the SCIAMACHY measurements, and vice-versa; both to confirm the simulation

studies, and better understand the sensitivity of the retrievals to differences in the radiance products. The same set of coincident SAGE II scans is used for this study, with comparisons performed in the same way as those presented in Section 3.

Figure 8 shows the USask retrieval applied to both instruments. Retrievals using the SCIAMACHY measurements agree very well with those using OSIRIS, and show many of the same biases with respect to SAGE II. Both instruments show underestimation with respect to SAGE II at high altitudes and latitudes. If this was due to inaccuracies in the assumed particle

size the error would be expected to change signs between hemispheres as the SCIAMACHY solar scattering angle goes from backscattering to forward scattering, which is not the case. Instead, these high altitude errors are more likely to be caused by errors in the assumed a priori extinction profile at high altitudes where the measurements are normalized, as the effect of this is nearly independent of solar geometry. From Figure 5 errors of $3 \times 10^{-6} \, \mathrm{km}^{-1}$ in the reference altitude range could explain biases of -30% at high altitudes. Additionally, both instruments have some stray light at these higher altitudes that increases the

radiance signal. This changes the shape of the aerosol measurement vector, and is likely a contributing factor to the low biases at high altitudes and latitudes. Unfortunately, both a priori and stray light errors have similar systematic biases on the profile making them difficult to separate except in simulation, and errors in the a priori can either help to cancel or exacerbate errors due to stray light. The shift in the SCIAMACHY measurements from low biases in the Southern Hemisphere to high biases in the Northern Hemisphere is present, as was seen in the IUP retrieval in Figure 1, again suggesting a particle size error. In

the USask retrieval this shift is approximately 20-30% between hemispheres, which from Figure 7 would be consistent with an overestimation of the Ångström coefficient by approximately 0.3, ie. an assumption of too large of particles at the high latitudes.





The IUP retrieval applied to both the SCIAMACHY and OSIRIS data is shown in Figure 9. OSIRIS solar scattering angles do not vary as strongly between the Northern and Southern hemispheres, and so the OSIRIS retrievals do not exhibit the same shift from low biases in the South to high biases in the North that are seen in the SCIAMACHY measurements. The impact of the a priori choice can also be seen here. For the OSIRIS retrievals the USask a priori was used without scaling, resulting in

low aerosol values in the normalization range and leading to lower aerosol values at all altitudes. However, if the IUP a priori is used the retrievals are substantially higher when compared to SAGE II (not shown). This is consistent with the results from section 4.2, in that larger a priori values in the normalization range lead to larger values at all altitudes.

This highlights the sensitivity to the chosen a priori and reference altitudes and the limitations of both the USask and IUP approaches. The USask technique of scaling an a priori profile that decays rapidly with altitude works with both instruments

provided the normalization altitude is chosen to minimize stray light. The variable normalization altitude ensures there is sufficient aerosol signal to determine the scaling, while the quickly decaying profile ensures the measurement vector is only weakly dependent on the scaling applied. However, while this provides a relatively robust retrieval it is likely to cause the aerosol to be underestimated at the normalization point, leading to low biases in the retrieved extinction, particularly at high altitudes. Conversely, the larger fixed a priori used in the IUP retrieval works well for SCIAMACHY when an appropriate

reference altitude is chosen, and can reduce biases at high altitudes. However, it yields poor results when applied to the OSIRIS measurements, illustrating the necessity of properly matching the normalization altitudes with the stray light characteristics and choice of a priori when using a fixed a priori profile.

## 6   Conclusions

The updated SCIAMACHY v1.4 aerosol extinction product shows good agreement with coincident SAGE II measurements,

typically within 20% for most regions. Exceptions to this include high Northern latitudes where larger positive biases of 20-40% are present, and altitudes above 25 km in the Southern high latitudes where negative biases are present. The differences between the limb and occultation measurements are well explained by two primary causes. First, the choice of a priori profiles is important in the limb retrieval due to the high altitude normalization. If the shape of the a priori profile is assumed incorrectly in the USask retrieval the scaling applied to the profile in the retrievals will produce incorrect aerosol in the reference altitude,

resulting in biases at all altitudes. The IUP retrieval fixes the aerosol profile above the retrieval range to the a priori value and errors couple similarly to lower altitudes. For both retrievals extinction errors in the reference altitude of $10^{-6} \, \text{km}^{-1}$ lead to errors in the retrieved extinction of 5% near the aerosol peak and up to 20% just below the reference altitude. Second, incorrect particle size generally shows a small mean difference when averaged over a range of scattering angles, but can have large differences of 100% or more for individual cases, particularly for strongly forward and backscattering viewing conditions. This

is especially important for orbits that systematically sample solar scattering conditions as a function of latitude. Simulations including a coarse mode of particles suggest a low bias in the retrieved extinctions during volcanically perturbed periods is likely for most geometries. However, the magnitude of the error is not expected to be systematically larger than the during background conditions on a profile-by-profile basis. Additionally, while the USask and IUP retrievals use the same particle



size assumptions, the biases are different for both the instruments and retrieval algorithms due to the difference in viewing geometries and definition of the measurement vectors. The error due to particle size can be reduced in backscatter geometries through the short-wavelength normalization. However, this normalization has the opposite effect in strongly forward scattering conditions, where it makes the retrievals more sensitive to particle size assumptions and measurement noise. Differences in

SASKTRAN and SCIATRAN radiative transfer models can cause systematic differences of up to 10% between the retrieved products, and may explain some of the vertical structure in the comparisons but are not expected to be a primary driver of the differences.

Future retrievals would benefit from improved a priori estimates of the aerosol extinction above 30 km, and particle size distributions. However, if this information remains limited, careful use of wavelength normalization (and the lack thereof) for

specific viewing geometries has the potential to reduce retrieval biases. Additionally, although the USask and IUP approaches to aerosol in the normalization range of the measurements are different (scaling vs. fixed to a priori respectively), both show comparable errors in the retrieved product for a given error in the normalization range. Robust measurements of high altitude aerosol are therefore needed to establish whether a fixed a priori or a scaled one leads to less error at these altitudes. In summary, this study investigates the retrieval of extinction from the limb viewing observations of scattered solar radiance by the satellite

borne instruments OSIRIS and SCIAMACHY. It provides a detailed analysis of our understanding of the systematic errors associated with these data products and biases with respect to the SAGE II measurements of extinction.

*Code and data availability.* Information on downloading the OSIRIS data set can be found at http://odin-osiris.usask.ca/?q=node/280. The SCIAMACHY data set can be downloaded at http://www.iup.uni-bremen.de/scia-arc/. SAGE data were obtained from the NASA Langley Research Center EOSDIS Distributed Active Archive Center. Information on downloading and using the SASKTRAN radiative transfer

model can be found at https://arg.usask.ca/docs/sasktran/ and the SCIATRAN code and documentation is available at http://www.iup.uni-bremen.de/sciatran/

*Acknowledgements.* This work was supported by the Natural Sciences and Engineering Research Council (Canada), the Canadian Space Agency, European Space Agency (ESA) through the SQWG and SPIN projects, the German Aerospace Center (DLR) through the SADOS project, the German Federal Ministry of Education and Research (BMBF) through the ROMIC project, the University and State of Bremen,

and the German Academic Exchange Service (DAAD) which provided a grant for Landon Rieger to visit the University of Bremen. Odin is a Swedish-led satellite project funded jointly by Sweden (SNSB), Canada (CSA), France (CNES), and Finland (Tekes).



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





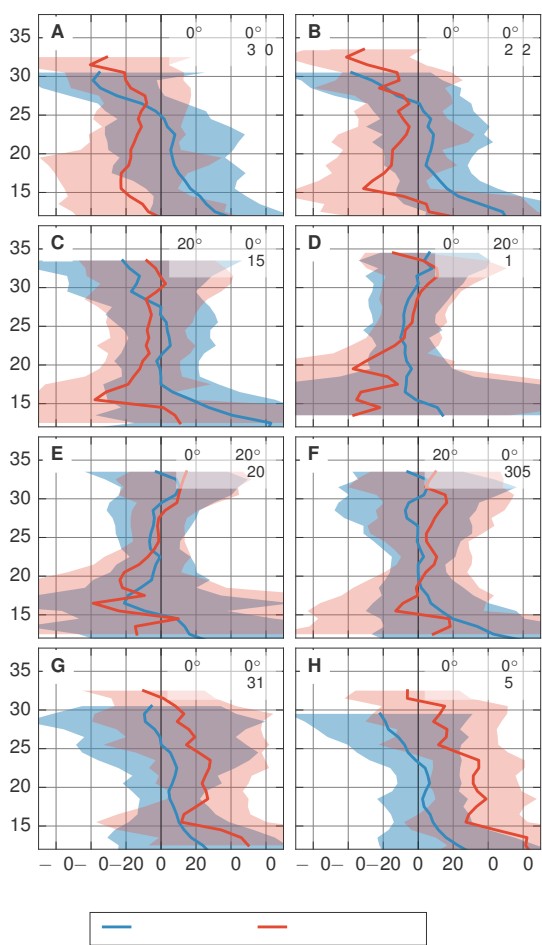

**Figure 1.** Coincident comparison between OSIRIS and SCIAMACHY measurements compared to SAGE II. Difference computed as (Instrument-SAGE II)/SAGE II × 100%. Shaded regions indicated one standard deviation of the differences from the median.





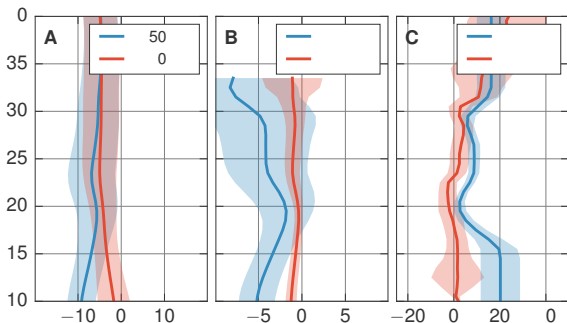

**Figure 2.** Comparisons of the radiative transfer models. Panel A shows the differences in radiance computed using SASKTRAN and SCIA-TRAN. Panel B shows the difference in measurement vectors. Panel C shows the difference in retrieved profiles. Differences in Panels A and B are computed as (SASKTRAN-SCIATRAN) / (SASKTRAN+SCIATRAN) × 200%. Extinction error is computed as (Retrieved - True) / True × 100%.

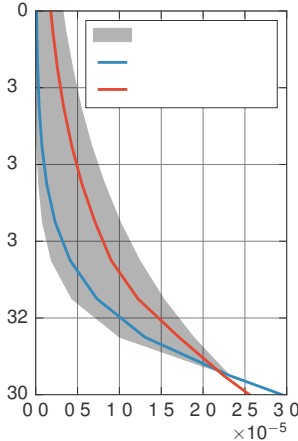

**Figure 3.** The range of the true state aerosol profiles is shown as the shaded region. The USask a priori is shown in blue and the IUP in red.



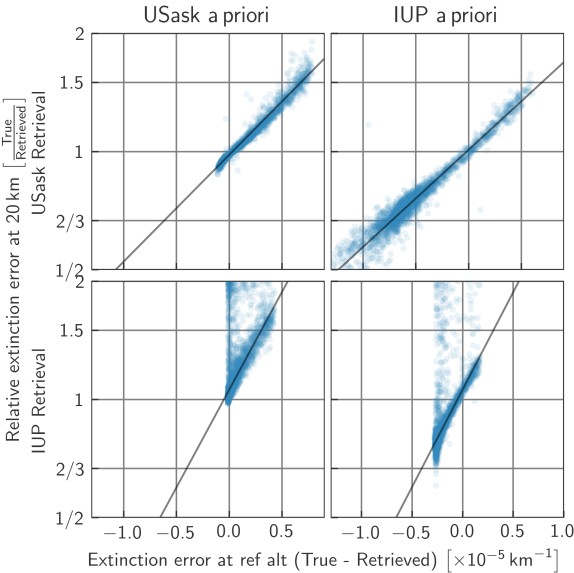

**Figure 4.** Relative error in the OSIRIS data retrievals at 20 km as a function of the absolute error in the true extinction at the reference point. The solid lines show the least squares fit to the data.

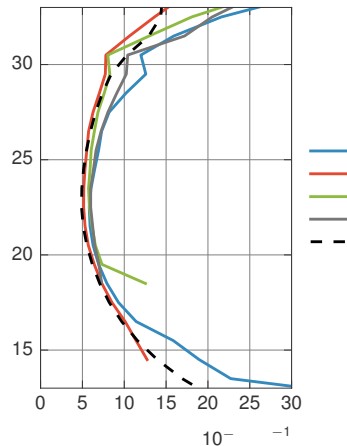

**Figure 5.** Percent error in the OSIRIS data retrievals as a function of altitude relative to an extinction error of $10^{-6}$ km$^{-1}$ at 35 km. Solid lines show values computed from the best fit line from the simulation studies shown in Figure 4. Dashed line shows the error expected from the linear error analysis of Equation 3.



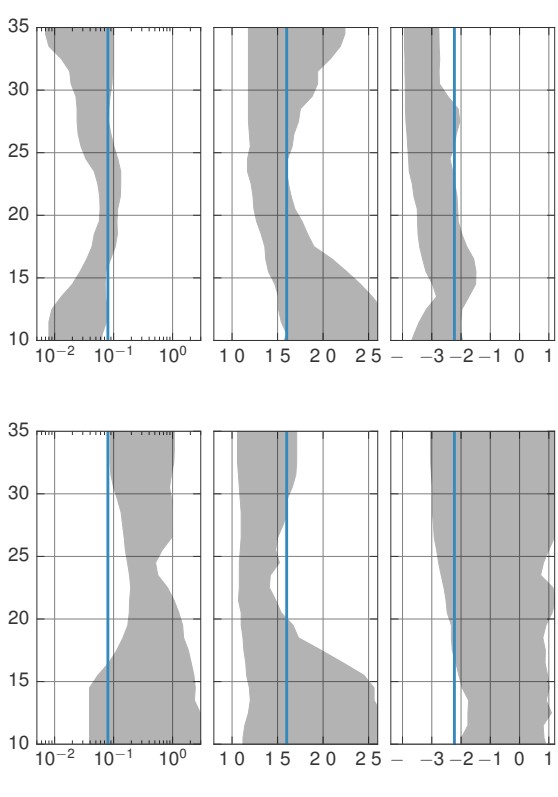

**Figure 6.** The range of particle sizes tested as a function of altitude. The top row shows the fine mode parameters, and the bottom row the coarse mode. The blue lines show the USask and IUP a priori values assumed in the retrievals. The grey shaded region shows the range of values used in the simulations.





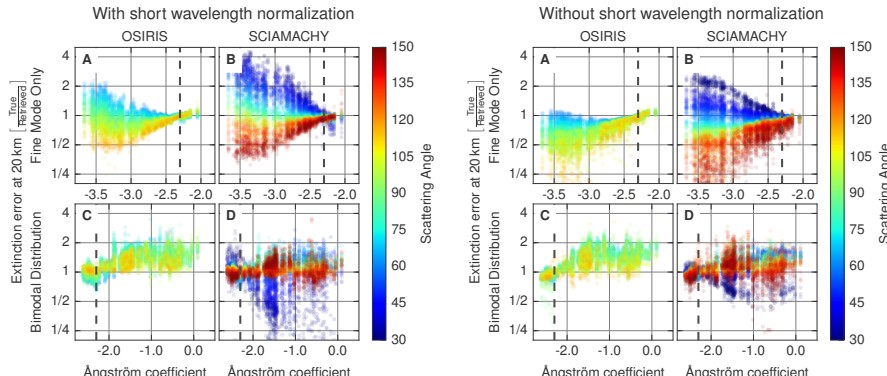

**Figure 7.** Error in the retrieved USask extinction as a function of Ångström coefficient at 20 km. The colour of the points show the solar scattering angle. The top row shows the error for conditions when only a fine mode of aerosol is present. The bottom row shows the error when there is both a fine and coarse mode distribution. The black dashed line indicates the Ångström coefficient corresponding to the particle size distribution used in the retrievals.



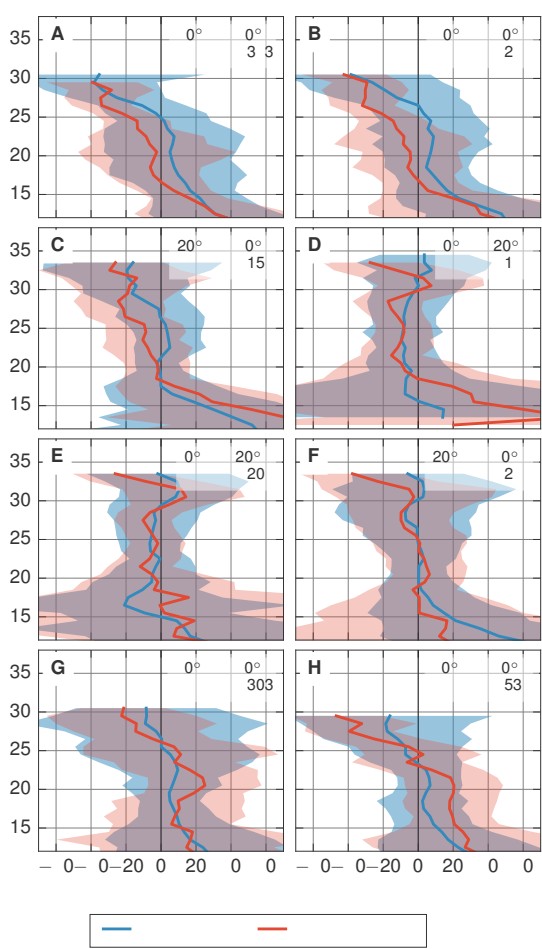

**Figure 8.** Coincident comparison with SAGE II when both OSIRIS and SCIAMACHY measurements have been processed with the USask algorithm.





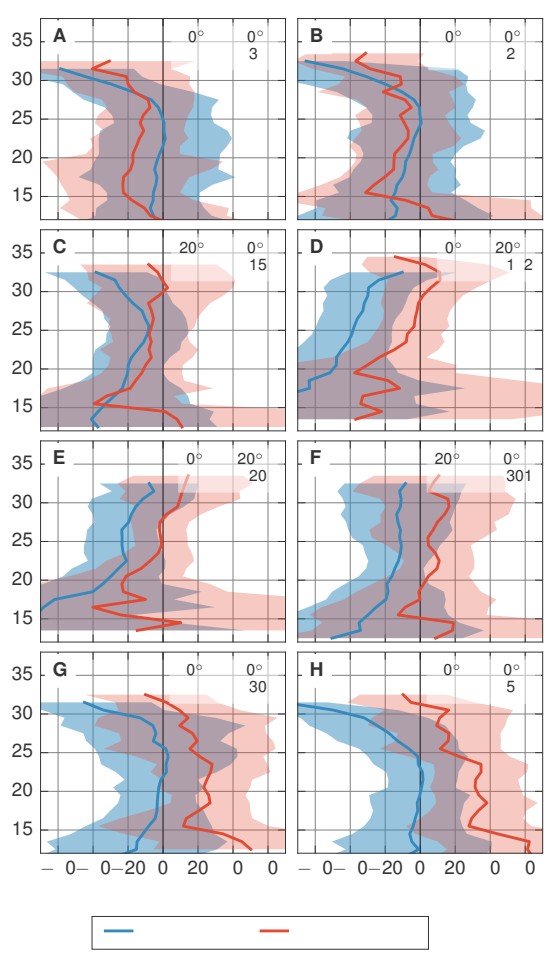

**Figure 9.** Coincident comparison with SAGE II when both OSIRIS and SCIAMACHY measurements have been processed with the IUP algorithm.