# Peer review of "A study of the approaches used to retrieve aerosol extinction, as applied to limb observations made by OSIRIS and SCIAMACHY"

_Atmospheric Measurement Techniques, 2017_

## Referee Comment (RC1) · Anonymous Referee #1 · 7 Feb 2018

General comments: This paper deals with aerosol retrieval algorithms applied to limb scattered solar radiation measured by OSIRIS and SCIAMACHY. The paper is well written and contains interesting and important observations for users of OSIRIS and SCIAMACHY aerosol data. One part of the paper, Sec. 4, is in my mind a little tedious reading if you are not expert user of aerosol data from these instruments. Anyhow, I would like to recommend the publication of this paper in AMT. I have the following comments and questions.

Specific comments: 1. p.3, Eq. (1): I don't find this equation from Bourassa (2012). Please explain and add general explanation why this odd complicated form. 2. p.3,

[Figure]

Eq.(1): Please explain N and m. 3. p.3, line 20: How do you determine where the aerosol contribution and stray light are at minimum, they both are unknowns? 4. p.3, line 18: At reference altitude, where do you get the reference atmosphere for the model vector? 5. p.4, Eq. (2): Please use the same ratio format as in Eq (1). 6. p.4, Eq. (2): Please explain what is the reason for these two very similar approaches (Eq. (1) and Eq.(2)) being still not identical? 7. p.4, Eq. (2): On line 11 you said that you drop the shorter wavelength normalisation, so what is lambda_ref in this equation? 8. p.4, line 18: Because your notation does not show that you normalise I by radiance from reference altitude, express this fact more clearly. Your I's are not measured physical quantities but ratios. Same applies to Eq. (1). 9. p.4, line 18: Same as above, what is your reference atmosphere? Is it same as with Eq. (1). 10. p.4, line 24: A constant S/N-ratio of 200, how is it possible? Radiance varies a lot as a function of altitude and wavelength. 11. p.5, line 2, SAGE was launched in 1984. 12. p 15, Fig.1: The legend box shows lines but not text Also axis labels are missing. These two things seem to be a problem for many other figs in this ms (only Fig. 7 looks OK). I have not been able to get a pdf of this ms. where figures are OK. 13. Sec. 4: I found this section quite tedious to read, that's why I have no specific comments except the following: 14. Sec. 4: Method: Use are using a method where you cross feed each intrument's retrieval system by data from other instrument's radiative transfer model. OK, perhaps this is a valid approach. I would like to see a more traditional approach as follows: 1) Agree about the atmospheric model or models (nor agreeing completely with a priori in retrievals), 2) Perform the radiative transfer calculation using Monte Carlo or other reasonably accurate scheme, 3) Make instrument simulations using OSIRIS and SCIAMACHY instrument simulation software (adding also noise), 4) Carry out retrievals for the synthetic data, 5) Compare results to the original atmospheric model. This ignores probably stray light but I do not know what to do with that. Is this approach too demanding for available time and resources? 15. p12, line12: "Robust measurements of high altitude aerosol. . .". What kind of instruments are you thinking about?

---

## Referee Comment (RC2) · Anonymous Referee #2 · 7 Feb 2018

General comments: The paper investigates sources and magnitude of various systematic differences between OSIRIS and SCIAMACHY aerosol retrievals as compared to SAGE II aerosol extinction coefficient measurements, using coincident measurements and simulations. ÂăIn an interesting approach, it also applies both algorithms to both datasets datasets in order to investigate the effect of assumptions made by each algorithm. The paper is well written and includes some interesting findings. I would like to recommend for publication subject to minor changes.

Specific comments:

1- Equation 1 and the description don't match, need to explain m and N. Are you using

a single or range of altitudes for normalization? If so, what is the range of altitudes used?

2- Section 2.2, can you briefly comment on the aerosol retrieval improvements of the updated V1.4 over the previous version?

3- Figure 7 and discussions are difficult to follow. Is the extinction error at different altitudes similar to 20km? I'd suggest plotting the comparison as profiles, for selected scattering angle and Angstrom coefficient range, or something similar and modify the text accordingly. The error due to particle size assumptions is very important and should be presented better.

4- Page 11, I find the discussion of this section and Fig 9 in particular lacking. Can you comment on IUP improvement of OSIRIS measurements (panels A B, G, H) compared to USask retrieval? Is it a result of using constant and higher normalization altitude used by the IUP retrieval?

5- Figure 2, the legend box needs to be moved to another position that doesn't interfere with the plot.

6- Page 12, last paragraph "Future retrievals would benefit from …." I'd like to see specific recommendations for each algorithm, rather than a general statement that all algorithms can benefit from.

---

## Referee Comment (RC3) · Anonymous Referee #3 · 9 Feb 2018

General comments on Rieger et al. (2018):

I look forward to reviewing a revised version of this paper. It is a well-defined study that addresses several important questions about limb scattering aerosol extinction retrievals, and makes useful recommendations for how they can be improved.

Unfortunately, the current version of this paper has a glaring deficiency: Most of the figures have been distorted at some point between their creation by the authors and their presentation in this journal. The fact that 7 of the 9 figures are in such poor condition makes the paper frequently tedious (and sometimes impossible) to review properly. Fixing this situation during the revision process will enable much more detailed analysis

of the work.

Specific comments:

Sect. 1, last paragraph:

Text should say "...a triple comparison among OSIRIS, SCIAMACHY and SAGE II..."

Sect. 2.1:

It would be useful to give the values of the assumed refractive index at the relevant wavelengths (470 and 750 nm).

The parameters $m$ and $N$ should also be defined more clearly.

Equation (1) also appears to contain an error: If $N$ = the number of altitudes averaged together to produce the altitude normalization, then the summation should cover $j_{ref} = m$ to $j_{ref} = m + N - -1$.

The next-to-last sentence of this section is also confusing: It states that "a modelled measurement vector assuming a molecular atmosphere... is also used as a normalization." Is this always done? If so, then why is it not included in equation (1)? And if not, how does the algorithm decide when it should be done or not done? And does one form of normalization replace the other, or are both used together? Finally, how does this change improve the convergence speed? (Some of these may be answered in another reference, so a citation may be all that's needed here.)

Sect. 2.2, 1st paragraph:

The fact that SCIAMACHY views the atmosphere in the "ram" direction (viewing direction aligned with satellite motion vector) should also be mentioned.

Sect. 2.2, 2nd paragraph:

Just to clarify: The v1.4 algorithm uses a fixed a priori extinction profile for altitudes outside the retrieval range, regardless of how large the difference between the a priori

and retrieved profile becomes? So the aerosol extinction profile used to simulate radiances by the radiative transfer model during the retrieval will contain (sometimes large) discontinuities at the edges of the retrieval range?

Sect. 2.2, 3rd paragraph:

Similar to the previous comment, it would be useful to give the assumed refractive index value at 750 nm.

Sect. 3, 1st paragraph:

The coincidence criteria are clearly stated and reasonable, but was an assessment of the resulting set of coincidences done to detect cases for which these criteria were met, but significant geophysical variability occurred between the 2 observations being compared? The relatively high sampling of the limb scattering measurements might make such an assessment possible, and it would be interesting to estimate how much of the differences between the occultation and scattering retrievals might result from true atmospheric variation (rather than deficiencies in either measurement).

Sects. 4 - 6:

This part of the paper contains many useful points, but it is difficult to evaluate the claims without better versions of the figures. Specifically:

Fig. 1 - Latitude ranges (upper right corner of each panel) are illegible, $x-$ and $y-$axes are not labeled, and legend (indicating the meaning of the line colors) is blank

Fig. 2 - Legend is again blank, $y-$axis numbers are garbled, and $x-$ and $y-$axes again are not labeled

Fig. 3 - Same problems as Fig. 2

Fig. 5 - Legend is again blank, and $x-$ and $y-$axes again are not labeled

Fig. 6 - Same problems as Fig. 5

Fig. 8 - Same problems as Fig. 1

Fig. 9 - Same problems as Fig. 1

---

## Author Comment (AC1) · 22 Feb 2018

The figures in the submitted manuscript do not display correctly for many readers. The attached file should correct this issue. My apologies to the reviewers that this caused problems for. While this is strictly an update to the figures, I look forward to addressing your other comments shortly, as well as any new ones that may arise.

Please also note the supplement to this comment:

[revised manuscript text omitted]

---

## Author Comment (AC2) · 11 Apr 2018

Thank you very much for the helpful comments and suggestions, attached is a detailed reply.

Please also note the supplement to this comment:
https://www.atmos-meas-tech-discuss.net/amt-2017-446/amt-2017-446-AC2-supplement.pdf

---

## Author Comment (AC4) · 11 Apr 2018

Thank you for the comments and recommendations, they have helped to clarify several important points.

Please also note the supplement to this comment:
https://www.atmos-meas-tech-discuss.net/amt-2017-446/amt-2017-446-AC4-supplement.pdf

---

## Author Response (AR1)

**1 Anonymous Review #1**

*p.3, Eq. (1): I dont find this equation from Bourassa (2012). Please explain and add general explanation why this odd complicated form.*

This is a simplified form of Eq's (5) and (7) of Bourassa et al., (2012). The Rayleigh normalization has been dropped, and the two equations combined to be directly comparable to the SCIAMACHY formalism shown later. The Rayleigh normalization is now discussed in more detail, including Eq. 2.
* * *
*p.3 Eq.(1): Please explain N and m.*

Explanation has been added.
* * *
*p.3, line 20: How do you determine where the aerosol contribution and stray light are at minimum, they both are unknowns?*

Thank you, this was poorly described in the manuscript. The description has been updated and the normalization made more explicit.
* * *
*p.3, line 18: At reference altitude, where do you get the reference atmosphere for the model vector?*

The atmosphere is from ECMWF interpolated to the scan location. This has been added to the manuscript on page 4 line 2.
* * *
*p.4, Eq. (2): Please use the same ratio format as in Eq (1)*

The SCIAMACHY measurement vector uses only a single wavelength, so there is no ratio.
* * *
*p.4, Eq. (2): Please explain what is the reason for these two very similar approaches (Eq. (1) and Eq.(2)) being still not identical?*

The SCIAMACHY approach shown in Eq. (2) (now Eq. 3) does not use a short-wavelength normalization.
* * *
*p.4, Eq. (2): On line 11 you said that you drop the shorter wavelength normalisation, so what is lambda ref in this equation?*

There is no lambda ref in Eq (2).
* * *
*p.4, line 18: Because your notation does not show that you normalise I by radiance from reference altitude, express this fact more clearly. Your Is are not measured physical quantities but ratios. Same applies to Eq. (1).*

The radiance, $I$, is a physical quantity. The normalization is performed by the second term in Eq. (1) and Eq. (2). (Eq(1) and Eq (3) in the updated manuscript)
* * *
*p.4, line 18: Same as above, what is your reference atmosphere? Is it same as with Eq. (1).*

Yes, the although the SCIAMACHY retrieval does not normalize the measurements by a rayleigh calculation, the retrieval uses the ECMWF atmosphere. This information is provided on Page 4, Lines 19-20.
* * *
*p.4, line 24: A constant/N-ratio of 200, how is it possible? Radiance varies a lot as a function of altitude and wavelength.*

Constant SNR is defined by the upper altitude normalization. In general SNR is decreasing from the lower to the upper altitudes, dividing the values at all the altitudes by the upper altitude radiance makes the latter one dominate in SNR.
* * *
*p.5, line 2, SAGE was launched in 1984.*

Thank you, corrected.
* * *
*p 15, Fig.1: The legend box shows lines but not text. Also axis labels are missing. These two things seem to be a problem for many other figs in this ms (only Fig. 7 looks OK). I have not been able to get a pdf of this ms. where figures are OK.*

Apologies for the figures, they should now be fixed in the updated manuscript.
* * *
*Sec. 4: I found this section quite tedious to read, thats why I have no specific comments except the following:*

If it's any consolation it was also a tedious, although we would argue necessary, piece of work to conduct.
* * *
*Sec. 4: Method: You are using a method where you cross feed each instruments retrieval system by data from other instruments radiative transfer model. OK, perhaps this is a valid approach. I would like to see a more traditional approach as follows:*

1. *Agree about the atmospheric model or models (nor agreeing completely with a priori in retrievals),*

2. *Perform the radiative transfer calculation using Monte Carlo or other reasonably accurate scheme,*

3. *Make instrument simulations using OSIRIS and SCIAMACHY instrument simulation software (adding also noise),*

4. *Carry out retrievals for the synthetic data,*

5. *Compare results to the original atmospheric model. This ignores probably stray light but I do not know what to do with that. Is this approach too demanding for available time and resources?*

We think this would be an interesting study as well, and has been done to a degree in the past (*Zawada et al., 2015*). However, in that case both models use much of the same internals such as quadrature methods, atmospheric setup, aerosol and other optical properties, etc. For this work, using an independent MC model to generate synthetic radiances would therefore be a test of these differences as well as a comparison between SASKTRAN and SCIATRAN. The approach used here eliminates the differences a third model would introduce.

*Zawada, D. J., et al. "High-resolution and Monte Carlo additions to the SASKTRAN radiative transfer mode" Atmospheric Measurement Techniques 8.6 (2015): 2609.*
* * *
*p12, line12: Robust measurements of high altitude aerosol. . .. What kind of instruments are you thinking about?*

Occultation would probably be the most likely, even if SAGE II measurements suffer from sensitivity to the retrieval process at these altitudes where molecular scattering and absorbing gases dominate the signal (see differences between v6.2 and v7).
* * *
**2   Anonymous Review #2**

*Equation 1 and the description dont match, need to explain m and N. Are you using a single or range of altitudes for normalization? If so, what is the range of altitudes used?*

Thank you, the text has been updated to include definitions of $M$ and $n$, and the description about the molecular normalization updated.
* * *
*Section 2.2, can you briefly comment on the aerosol retrieval improvements of the updated V1.4 over the previous version?*

We havent quantitatively assessed how much better V1.4 of algorithm performs in comparison to V1.1. However, the qualitative improvements are:

1. The overall uncertainty of the results is reduced by dropping the 470 nm normalization (as the normalization introduces additional noise associated with the measurement at 470 nm).

2. The albedo retrieval enables us to better parameterize the underlying surface in comparison to the database albedo values used in V1.1.

3. The retrieval is done on the measurement grid, which allows us to minimize the need for constraints and to avoid additional errors, e.g. related to altitude interpolation.

This information has been added on Page 5 lines 14-16.
* * *
*Figure 7 and discussions are difficult to follow. Is the extinction error at different altitudes similar to 20km? Id suggest plotting the comparison as profiles, for selected scattering angle and Angstrom coefficient range, or something similar and modify the text accordingly. The error due to particle size assumptions is very important and should be presented better.*

Thank you, it is a bit tricky to show so much information on one plot and this is a nice suggestion. It is difficult to group by the Ångström coefficient while showing altitude dependence as the particle size changes with altitude, so we have used the line colouring to indicate this. The additional plot (now figure 7) has been added to the manuscript with additional explanation and hopefully this clarifies the discussion. We think Figure 7 is still worth including in the paper, as it helps show that for a unimodal case the error is well categorized by the Ångström coefficient, but not for a bimodal case.
* * *
*Page 11, I find the discussion of this section and Fig 9 in particular lacking. Can you comment on IUP improvement of OSIRIS measurements (panels A B, G, H) compared to USask retrieval? Is it a result of using constant and higher normalization altitude used by the IUP retrieval?*

Thank you, this is an important point. It is a result of using a different a priori at a different normalization altitude which will also have a different amount of stray light. However, the relative contribution of each of these is unknown, as the true high altitude profile and stray light is not known. A more detailed description been added to the manuscript on Page 12 lines 16-19.
* * *
*Figure 2, the legend box needs to be moved to another position that doesn't interfere with the plot.*

Plot has been updated.
* * *
*Page 12, last paragraph Future retrievals would benefit from . . .. Id like to see specific recommendations for each algorithm, rather than a general statement that all algorithms can benefit from.*

More specific recommendations have been added on page 13 lines 11-13

**3 Anonymous Review #3**

*Sect. 1, last paragraph: Text should say ...a triple comparison among OSIRIS, SCIAMACHY and SAGE II...*

Updated
* * *
*Sect. 2.1: It would be useful to give the values of the assumed refractive index at the relevant wavelengths (470 and 750 nm).*

This has been added on page 3, line 16.
* * *
*The parameters m and N should also be defined more clearly.*

Definition of these parameters has been added to the text.
* * *
*Equation (1) also appears to contain an error: If N = the number of altitudes averaged together to produce the altitude normalization, then the summation should cover jref = m to jref = m + N  1.*

Thank you, Eq 1 has been updated.
* * *
*The next-to-last sentence of this section is also confusing: It states that a modelled measurement vector assuming a molecular atmosphere... is also used as a normalization. Is this always done? If so, then why is it not included in equation (1)? And if not, how does the algorithm decide when it should be done or not done? And does one form of normalization replace the other, or are both used together?*

Thank you, this was poorly explained. The Rayleigh normalization is always used in addition to the altitude and wavelength normalization. The normalizations are now discussed in more detail on page 3 lines 20-30
* * *
*Finally, how does this change improve the convergence speed? (Some of these may be answered in another reference, so a citation may be all thats needed here.)*

The convergence of Chahine relaxation techniques is discussed by *Chu, 1985* and *Barcilon, 1975*, and a reference has been added to manuscript on page 3 line 21. However, briefly, they show that for a measurement, $y$ at altitude $j$ given by

$$y_j = \sum_i K_{ij} x_i$$

where $K$ is the kernel and $x$ the state vector, the convergence is given by

$$\hat{\mathbf{x}}^{(n+1)} - \mathbf{x} = \mathbf{Q}(\hat{\mathbf{x}}^{(n+1)} - \mathbf{x})$$

where

$$\mathbf{Q} = \mathbf{I} - \mathbf{z}\mathbf{K}$$

Where $\mathbf{I}$ is the identity matrix and $\mathbf{z}$ is a diagonal matrix with elements

$$z_{jj} = \frac{x_j^{(n)}}{\sum_i K_{ij} x_i^{(n)}}$$

The speed of convergence is dictated by the matrix norm of $\mathbf{Q}$. If the measurement vector $y$ includes a positive offset $a$, such as one due to Rayleigh scattering,

$$y_j = \sum_i K_{ij} x_i + a_j$$

Then the elements of $z$ become

$$z_{jj} = \frac{x^{(n)}}{\sum_i K_{ij} x_i^{(n)} + a_j}$$

This reduces $z$ and increases the norm of $\mathbf{Q}$, slowing convergence. However, perhaps a more intuitive approach is simply to look at a one dimensional problem:

$$y = Kx + a$$

$$\hat{x}^{(n+1)} = \hat{x}^{(n)} \frac{y}{F} = \hat{x}^{(n)} \frac{Kx + a}{K\hat{x}^{(n)} + a} = x + \frac{a(\hat{x}^{(n)} - x)}{\hat{x}^{(n)} K} - \frac{a^2(\hat{x}^{(n)} - x)}{(\hat{x}^{(n)})^2 K^2} + \cdots$$

As $a$ increases from zero, the solution is damped towards the current state. As other minimization methods generally use the kernel $K$ directly, instead of relying on the approximation of $y/F$, this subtraction of the Rayleigh background is not required in those cases (at least for convergence purposes).
* * *
*Sect. 2.2, 1st paragraph: The fact that SCIAMACHY views the atmosphere in the "ram" direction (viewing direction aligned with satellite motion vector) should also be mentioned.*

This information has been added to page 4 lines 7-8.
* * *
*Sect. 2.2, 2nd paragraph: Just to clarify: The v1.4 algorithm uses a fixed a priori extinction profile for altitudes outside the retrieval range, regardless of how large the difference between the a priori and retrieved profile becomes? So the aerosol extinction profile used to simulate radiances by the radiative transfer model during the retrieval will contain (sometimes large) discontinuities at the edges of the retrieval range?*

Correct. Although results from Figure 4 and 5 suggest that it is really the magnitude of the error in the normalization range that is the main driver of error at lower altitudes, and not discontinuities in the profile. As scaling the aerosol values in the normalization range based on lower altitude retrieved values impacts the entire profile, which in turn impacts the normalization range, it is not clear what approach should be favoured.
* * *
*Sect. 2.2, 3rd paragraph: Similar to the previous comment, it would be useful to give the assumed refractive index value at 750 nm.*

This has been added on page 5 line 9.
* * *
*Sect. 3, 1st paragraph: The coincidence criteria are clearly stated and reasonable, but was an assessment of the resulting set of coincidences done to detect cases for which these criteria were met, but significant geophysical variability occurred between the 2 observations being compared? The relatively high sampling of the limb scattering measurements might make such an assessment possible, and it would be interesting to estimate how much of the differences between the occultation and scattering retrievals might result from true atmospheric variation (rather than deficiencies in either measurement).*

Previous studies have used considerably tighter coincident criteria of $1°$ latitude, $1000\,\text{km}$ distance and $\pm 24$ hours when comparing OSIRIS and SAGE II (*Rieger et al., 2015*). These comparisons showed approximately the same magnitude and patterns of biases as the looser criteria used in this work. Therefore, tightening the criteria is not expected to reduce natural variability significantly compared to measurement variability, while greatly reducing the number of triple coincidences. This is at least partially due to the scanning nature of the instruments that often take a few degrees latitude to complete a vertical profile, making it difficult to more precisely define a 'coincident' measurement. A note about this has been added to the manuscript on page 5 lines 26-27.

To fully explore the natural vs measurement variability and take advantage of the high sampling would likely require a much more detailed study, where each measurement is treated not as part of a profile, but as an independent sample at the tangent point location (likely on equivalent latitude). At least in the case of scanning instruments such as OSIRIS, SCIA, and SAGE; for OMPS-LP, where an image is taken this effect may not be as important.

*Rieger, L. A., A. E. Bourassa, and D. A. Degenstein. "Merging the OSIRIS and SAGE II stratospheric aerosol records." Journal of Geophysical Research: Atmospheres 120.17 (2015): 8890-8904.*
* * *
*Sects. 4 - 6: This part of the paper contains many useful points, but it is difficult to evaluate the claims without better versions of the figures. Specifically:*

- *Fig. 1 - Latitude ranges (upper right corner of each panel) are illegible, x and yaxes are not labeled, and legend (indicating the meaning of the line colors) is blank*

- *Fig. 2 - Legend is again blank, yaxis numbers are garbled, and x and yaxes again are not labeled*

- *Fig. 3 - Same problems as Fig. 2*

- *Fig. 5 - Legend is again blank, and x and yaxes again are not labeled*

- *Fig. 6 - Same problems as Fig. 5*

- *Fig. 8 - Same problems as Fig. 1*

- *Fig. 9 - Same problems as Fig. 1*

Apologies for the figure errors. It seems to be an embedded font issue that is now fixed.
* * *

[revised manuscript text omitted]